# Effect of Abrasive Grain Concession on Micromechanical Behavior of Lapping Sapphire by FAP

**DOI:** 10.3390/mi13081322

**Published:** 2022-08-16

**Authors:** Huimin Xu, Jianbin Wang, Yiliang Xu, Qingan Li, Benchi Jiang

**Affiliations:** School of Mechanical Engineering, Anhui Polytechnic University, Wuhu 241000, China

**Keywords:** sapphire, molecular dynamics, lapping, cutting depth, removal mechanism

## Abstract

Aiming at exploring the material removal mechanism for sapphire using diamond abrasive grains at the microscopic level, this paper modeled and analyzed the microscopic yield behavior of diamond abrasive grains in the FAP grinding process of sapphire. Molecular dynamics were used to simulate the effects of abrasive particle size on the cutting force, potential energy, and temperature in the Newtonian zone during micro-cutting. The effect of different abrasive particle sizes on material removal was analyzed through experiments. The simulation results show that the abrasive particle radius was 12 Å, the micro-cutting force reached more than 3500 nN, while the cutting force with an abrasive particle radius of 8 Å only reached 1000 nN. Moreover, the potential energy, cutting force, and temperature in the Newtonian zone between the sapphire crystal atoms also increased. The results showed that the material removal rate saw a nonlinear increasing trend with the increase in particle sizes, while the surface roughness showed an approximately linear increase. Both of them showed a similar trend. The experimental results lay a theoretical basis for the selection of the lapping process parameters in sapphire.

## 1. Introduction

Sapphire is typically a hard and brittle material with excellent physical and chemical properties and is widely used in civil, military, aerospace, and other fields. The chemical composition of single-crystal sapphire is alumina (a-Al_2_O_3_). Its constituent elements Al^3+^ and O^2−^ belong to the typical hexagonal close-packed arrangement. Al^3+^ ions occupy about two-thirds of the inter-octahedral gap created between the densely packed O^2−^. Three O^2−^ ions in the sapphire molecule are in the same layer. The Al^3+^ ions are alternately arranged in pairs with the O^2−^ ion layer, and repeats every three layers, keeping the ratio of coordination numbers to O^2−^ ions at 2:3. Due to the stable coordination polyhedral structure, the cleavage and slip of a single-crystal sapphire at room temperature are very difficult to activate.

However, the performance and life of sapphire devices are restricted by the quality of their surface processing [1,2,3]. Lapping is the main process to obtain a high-quality surface for sapphire. The traditional material removal method uses free abrasive slurry. During the process, the free abrasive rolls on the surface of the workpiece produce a rolling effect, which can easily cause damage to the surface and subsurface of the workpiece, and the excessive waste of abrasives can easily cause environmental pollution. Because fixed abrasive grains touch the convex part of the workpiece surface, high-quality surface can be obtained with a few abrasive removals, which has obvious advantages in the sapphire lapping process [4,5].

During the fixed abrasive pad (FAP) process, due to the normal force and gravity of diamond abrasive grains, the soft matrix material “gives” easily, which leads to changes in the penetration depth of the abrasive grains and affects the material removal. Wang et al. [6] studied the effects of different cutting depths on the subsurface damage and material removal of Al-Si double layers. The results showed that the chip volume increased as the cutting depth increased, which not only led to a higher temperature in the workpiece, but also a greater lapping force was required to maintain a stable lapping process. Li et al. [7] studied the influence of different cutting depths on the mechanical properties of the workpiece, and the results showed that the number of lattice transformations increased with the increase in the cutting depth in the nano-cutting process. This is consistent with the evolution of microscopic defects. Zhou et al. [8] found that as the cutting depth of nano-abrasives on the surface of SiC workpieces increased, the number of phase change atoms caused by cutting and wear increased. The greater the cutting depth of the abrasives is, the deeper the subsurface damage of the workpiece is, and more material is removed. Wu et al. [9] studied that the grinding particle size can be optimized to improve the grinding surface quality. Chen et al. [10] studied the influence of cutting edge radius on material removal in micro milling.

As the size of the abrasive particles becomes smaller, the cutting depth becomes smaller. The disadvantage of the existing detection techniques is that the removal behavior of abrasive particles at the micro-scale cannot be directly observed. However, this subtle process can be better studied by molecular dynamics simulation [11]. Some researchers have used molecular dynamics simulation to study the processing of hard and brittle materials at the microscopic scale [12,13,14]. In the micro-scale machining simulation process, the molecular dynamics simulation method is not only effective and accurate, but it shows intuitiveness in the evolution of the atomic scale [15]. Dai et al. [16] studied the process of polishing silicon with diamond abrasive grains, and the results showed that the increase in the polishing depth increased the phase change in the silicon, the temperature, and the potential energy. Using molecular dynamics simulation, Zhao et al. [17] found that the ultrasonic vibration of the cutting tool promoted ductile mode cutting, and a larger cutting depth promoted the phase transition of a cubic diamond. Zhao et al. [18] used molecular dynamics to study the effect of nanoparticles on nano-cutting of single crystal materials. The study indicated that through molecular dynamics simulations, the quality of the processed surface can be improved by adjusting the cutting depth. Xie et al. [19] found that when the cutting depth was reduced to the size of the atom or below the size of the atom, the material was removed by the formation of chips through dislocation movement. Dai et al. [20] used molecular dynamics simulation to establish a model for polishing single-crystal silicon with diamond abrasive grains. The results showed that as the polishing depth increased, the cutting force, the depth of the subsurface damage, and the potential energy increased.

Molecular dynamics (MD) simulation was proven to be a strong candidate for studying the mechanism of FAP removal at the nanoscale [21]. However, there are few literature reports on the material removal and surface forming mechanism of FAP grinding sapphire micro-cutting by using molecular dynamics simulation. Therefore, from the perspective of pure mechanical action, this paper modeled and analyzed the diamond abrasive particle size and microscopic yield behavior during the grinding of sapphire on an FAP. We discuss the influence of the yield behavior on the two removal forms in the process of sapphire material removal. In addition, this paper analyzed the influence of different abrasive particle sizes on material removal and used molecular dynamics to simulate the influence of different abrasive particle sizes on the cutting force, potential energy, and temperature in the Newtonian zone after abrasive grain yield, which will help transform the empirical knowledge of sapphire micro-cutting technology into theoretical knowledge and provide theoretical support for sapphire chemical mechanical polishing.

## 2. Analysis of the Yield Model

In order to facilitate the analysis and simplify the model, the following assumptions were made: the diamond abrasive particles were set as spheres, the abrasive grains were independent and evenly distributed on the fixed abrasive pad, and the particle size and shape were consistent. There was a gap between the sapphire and the FAP.

As shown in Figure 1, when the sapphire workpiece is in contact with the surface of the FAP, it is transferred to the surface of the workpiece under the action of the load F, and the contact surface of the polishing pad and the workpiece are subjected to a normal force, which realizes the height of abrasive grains with a higher exposure height yield to equal with that in the abrasive grains with a lower exposure height, thus achieving super smooth and damage-free polishing of the surface of the workpiece. Therefore, it was necessary to analyze the plastic intrusion of diamond abrasive grains in the workpiece and the elastic retreat characteristics of the matrix in the FAP.

Diamond abrasive grains are pressed into the FAP under the action of normal force F so that the base material with which it is in contact is elastically deformed. Its elastic deformation amount$\;\delta_{P}$ is the distance that diamond abrasive grains yield in the FAP. The relationship between the normal force F and the normal displacement it produces is shown in formulas (1) and (2):(1)$$F=\frac{4}{3}E^{*}{\left(\frac{D}{2}\right)}^{\frac{1}{2}}\delta_{P}{}^{\frac{3}{2}}$$
(2)$$E^{*}=\frac{E_{p}}{1-v_{p}{}^{2}}$$
where F is the normal force; D is the diameter of the diamond abrasive grains; in um, $E^{*}$ is the elastic modulus of the abrasive grains; in GPa, $E_{p}$ is Young’s modulus of the matrix; and in GPa, $v_{p}$ is the matrix Poisson’s ratio. The retreat distance of the diamond abrasive grains in an FAP is related to the elastic modulus, particle size, and normal force of the abrasive grains. Under the same conditions of other parameters, the smaller the particle size of the abrasive particles, the greater the retreat distance of the diamond abrasive particles is in the polishing pad.

If the polishing pad substrate is hard without considering the “yield” of the abrasive grains, $d_{1}$ is the cutting depth of the abrasive grains pressed into the sapphire, $d_{2}$ is the depth of the abrasive grains embedded in the polishing pad, and D is the abrasive grain size. The lapping pressure is mainly shared by the abrasive grains with a larger exposed height, so the lapping load of a single abrasive grain is larger, which leads to a corresponding increase in the depth of its cutting into the workpiece. Due to the “recession”, the depth of the retreat is $\delta_{P}$, and the cutting depth of the abrasive grains pressed into the sapphire after the retreat is $d_{1}-\delta_{p}$. Thus, the depth of cut decreases after the retreat. During the cutting process, the energy generated by the plastic deformation of the sapphire crystal resisting the extrusion of the abrasive grain is reduced, making it easier to form ductile removal as far as the form of material removal is concerned. When the depth of cut is reduced to less than a certain critical value, the energy generated by the plastic deformation is not enough for the brittle rupture of the crystal material, and the brittle rupture crack extension is effectively controlled, and the material removal will be in a ductile removal mode. Therefore, the cutting depth of abrasive particles is an important basis to determine whether brittle hard materials can be carried out in the ductile removal mode. The ductile removal mode of a material is often used as a way to significantly improve the surface quality of machining. For sapphire lapping, controlling the depth of cut after yielding affects the crack spreading and the surface quality of the sapphire. In addition, during the removal of sapphire material, the shaded area of the triangle, as shown in Figure 1, increases with the increase in the depth of cut of the abrasive grains after retreating. The triangular shaded area is the cross-sectional area of the abrasive grain cutting and plowing of the sapphire. When the linear speed of the abrasive grain is constant, the greater the cutting depth is, the larger the plowing area is, and more volume of the material is removed, so the brittle removal mode will effectively improve the removal rate of sapphire material.

S is the cross-sectional area of the abrasive grains cut into the sapphire; r is the radius of the section circle of the depth of penetration of the spherical diamond grit; $\delta_{ave}$ is the average depth of cut. Since the exposed height of the diamond abrasive grains on the FAP surface is uniformly distributed, the depth of the sapphire workpiece is also uniformly distributed under the action of the load. Therefore, the average depth of cut of diamond abrasive grains to the sapphire workpiece should be half of the maximum depth of cut.

Because the depth of diamond abrasive cutting into the workpiece is much smaller than its diameter, the actual cut-in cross-sectional area should be the difference between the fan-shaped area and the triangular area in Figure 1.
(3)$$\theta =arcos\frac{D/2-\delta_{ave}}{D/2}$$
(4)$$r=\sqrt{{\left(\frac{D}{2}\right)}^{2}-{\left(\frac{D}{2}-\delta_{ave}\right)}^{2}}=\sqrt{\delta_{ave}\times D-\delta_{ave}{}^{2}}\approx \sqrt{\delta_{ave}D}$$
(5)$$S=\frac{2\theta }{360}\times \pi \times {\left(\frac{D}{2}\right)}^{2}-\frac{1}{2}\times 2r\times \left(\frac{D}{2}-\delta_{ave}\right)$$

After substituting formulas (3) and (4) into formula (5), the actual cross-sectional area of a single diamond abrasive grain cutting into the surface of the workpiece is about:(6)$$S\approx \frac{1}{180}\times \pi \times {\left(\frac{D}{2}\right)}^{2}\times arcos\frac{D/2-\delta_{ave}}{D/2}-\left(\frac{D}{2}-\delta_{ave}\right)\times \sqrt{\delta_{ave}D}$$
where the grinding time of a single diamond abrasive particle is t, and the material removal amount ΔQ is:(7)ΔQ=K×S×ν×t

K represents the ratio of the actual material removal amount to the theoretical removal amount during processing, which is generally a constant of less than 1, and ν is the movement speed of a single diamond abrasive grain. Thus, the material removal rate (MRR) of a fixed abrasive grinding sapphire workpiece is:(8)$$MRR=\frac{\Delta Q\times N_{m}}{t}$$
(9)$$N_{m}=\frac{\Lambda X_{s}\xi S_{w}}{500\pi D^{2}}$$

$N_{m}$ is the actual number of diamond abrasive grains in contact with the workpiece on the corresponding area of the workpiece; Λ is the ratio of the protruding area of the FAP pad to the total area; $X_{s}$ is the area fraction of diamond abrasive particles on the surface of the fixed pad; ξ is the number of abrasive particles actually contacting the workpiece; $S_{w}$ is the surface area of the workpiece.

Substituting formulas (6), (7), and (9) into formula (8), the material removal rate of sapphire with FAP grinding can be obtained as:(10)$$MRR=\frac{\Lambda X_{s}\xi K\nu S_{w}}{500\pi D^{2}}\times \left[\frac{1}{180}\times \pi \times {\left(\frac{D}{2}\right)}^{2}\times arcos\frac{D/2-\delta_{ave}}{D/2}-\left(\frac{D}{2}-\delta_{ave}\right)\times \sqrt{\delta_{ave}D}\right]$$

In formula (10), when working with fixed abrasives, the parameters that affect the material removal rate include the rotation speed of the grinding disc, the volume concentration of the diamond abrasive in the FAP, the linear correlation of the number of abrasive grain groups in contact with the workpiece, the particle size of the diamond abrasive and the depth of the abrasive grain cutting, etc. Therefore, in order to improve the material removal rate, in addition to optimizing the grinding process parameters, it is also necessary to continuously improve the structure of the FAP, and appropriately control the particle size of the abrasive grains to increase the depth of its cutting into the workpiece.

In order to measure the surface roughness of the sapphire workpiece after fixed abrasive grinding, we defined the surface contour curve of the workpiece as the function f(x). Within a certain sampling length L, and assuming that the number of peaks and valleys is roughly equal, the deepest valley bottoms and plastic uplift peaks should occur at the maximum cutting depth of the abrasive grains. Since the exposed height of the abrasive particles is consistent with the surface contour of the substrate, the average peak valley height of the workpiece surface contour can be approximately regarded as the average depth of cut of the abrasive particles. Thus, the arithmetic mean of the heights of each point on the surface contour of the workpiece can be obtained, namely the surface roughness (*Ra*), as shown in Figure 2.
(11)$$R_{a}=\frac{1}{L}\int_{0}^{L}\left|f\left(x\right)\right|\varphi dx=\frac{1}{4}\delta_{ave}tan\frac{\varphi }{2}$$

Here, φ is the included angle of the cross-section of the diamond abrasive grains cutting into the sapphire workpiece. It has been assumed that the diamond abrasive grains have a spherical structure, so the tangent function of the included angle of the section can be obtained:(12)$$tan\frac{\varphi }{2}=\frac{r}{\delta_{ave}}=\frac{\sqrt{\delta_{ave}D}}{\delta_{ave}}$$

The surface roughness (*Ra*) of a sapphire workpiece ground by fixed abrasives can be calculated as:(13)$$R_{a}=\frac{1}{4}\sqrt{\delta_{ave}D}=\frac{1}{8}\sqrt{2D\delta_{max}}$$

It can be seen in formula (13) that the particle size of the diamond abrasive in the FAP and the maximum depth of its cutting into the workpiece are the main parameters of the surface roughness value. The surface of the workpiece is left with incisions due to the exposed diamond abrasive particles, causing fluctuations in the surface morphology. Among them, the FAP contains large-diameter diamond abrasives and has surface abrasive particles with a large, exposed height, and the depth of the cutting into the workpiece increases accordingly. The residual cutting marks on the surface of the workpiece are deep, so the surface roughness value is large. Therefore, in order to improve the surface processing quality of the workpiece, the particle size of the diamond abrasive in the FAP should be appropriately reduced, and the depth of its cutting should be controlled. 

## 3. Molecular Dynamics Simulation

With the development of manufacturing technology, the microscopic removal of materials from the wafer surface is at the nanoscale/sub-nanoscale [22,23,24]. However, molecular dynamics simulation can reveal the position of each atom, the cutting force, the evolution of temperature, and the potential energy during the cutting process [25]. Moreover, it can analyze the influence of different cutting depths caused by abrasive grain withdrawal on sapphire processing.

Due to the special lattice structure of single-crystal sapphire, the bond length and the bond energy between ions are obviously different, resulting in the difference between the energies of each face, which is typical anisotropy. The crystal planes known so far are: c (0001), r (10$\bar{1}$1), a ($11\bar{2}0$), m ($10\bar{1}0$), n ($22\bar{4}3$), p ($11\bar{2}3$), R ($01\bar{1}2$), s ($0\bar{2}21$), etc. As shown in Figure 3 [26], the different faces have different angles. Among them, the surface energy of the c-plane of the single-crystal sapphire is relatively low, and the equivalent coefficient of slip between the crystal basal planes is low. It is easier to achieve a certain plastic deformation through slip. At present, regarding sapphire crystal orientation, the c-plane is the most widely used [27]. The sapphire workpieces discussed in this article all refer to single-crystal sapphire with face c (0001). 

In this study, open-source software was used for the molecular dynamics simulation (LAMMPS), and the simulation data were processed by the visualization Ovito software. (Version: 3.44, which was developed by a team of scientists at the Technical University of Darmstadt, Hesse, Germany) The choice of potential function is the main factor for the accuracy of molecular dynamics simulation. The Matsui potential is the closest to the lattice constant, elastic constant, and surface energy of α-Al_2_O_3_ using experimental simulation. Therefore, the Matsui potential function was selected as the atomic potential of sapphire [28]. It built a model through the lattice custom command in the LAMMPS custom modeling method. The lattice constant of single-crystal alumina was entered (a = 4.78 Å, c = 12.9914 Å). The length of the basis vector was determined according to the crystallographic information parameters of alumina. Then, the atomic coordinates of Al and O were entered. The atoms were filled into the box. The relative atomic masses of Al and O were defined, and then the atomic model was established. The temperature of the constant temperature zone was controlled at 300 K, and the time step of the simulation was chosen as 1 fs, with 1000 steps of relaxation under the NVT system. Periodic boundary conditions were used, and free boundary conditions were used for the remaining directions. The particle size was 8 Å, 10 Å, 12 Å, the micro-cutting depth was 5 Å, the velocity direction was x-positive, and the velocity was 150 m/s. In this study, the Verlet algorithm was chosen due to its concise execution and lower memory requirement. Table 1 shows the specific parameters of the model.

As shown in Figure 4, the model was composed of a rigid diamond abrasive grain and a sapphire workpiece. The size of the workpiece model in the x, y, and z directions was 90 Å × 30 Å × 40 Å, containing 22,004 atoms. The diamond grain was a hemispherical model with 2374 atoms. Since the hardness of the diamond (Mohs 10) is greater than that of sapphire (Mohs 9), the diamond grain was set as a rigid body without considering the wear and deformation of the diamond in cutting. Similar to previous molecular dynamics simulations, the workpiece matrix was divided into three sections: the boundary zone, the constant temperature zone, and the Newtonian zone (processing area). During the simulation, in order to reduce boundary effects, atoms were not involved in motion in the boundary zone and remain fixed in space. The constant temperature zone plays the role of reasonable heat dissipation and temperature stabilization [29]. The atomic motion of the Newtonian zone and the constant temperature zone obeys Newton’s second law [30].

## 4. Results and Analysis

### 4.1. Analysis of the Simulation Results

Figure 5 shows the simulation diagram of the abrasive micro-cutting process with three different radii. In order to better reflect the micro-cutting state of sapphire workpieces under the action of different particle sizes, the main view and top view were selected for analysis. In comparing the simulation results, as the radius of the abrasive grain was larger, a larger atomic volume of chips accumulated in front of the abrasive grain. Moreover, the workpiece atoms were obviously extruded and displaced with the increase in the abrasive radius. The widths of both sides of the workpiece were the widest when the abrasive radius was 12 Å and the material was finally removed.

As shown in Figure 6a, it can be seen that the micro-cutting forces of the three radii show states of first increasing, then decreasing, and then reaching a stable fluctuation. The micro-cutting force with a radius of 12 Å was the first to reach the maximum, and when the time step was about 7000, the micro-cutting force reached 4000 nN. In addition, it can be seen that with the increases in particle size, the steady-state micro-cutting force increased accordingly to 700 nN, 1000 nN, and 1250 nN, respectively. According to the analysis, the larger the abrasive particles were, the larger the contact area was between the abrasive particle and the workpiece during cutting. Moreover, the more atomic bonds were destroyed by the abrasive particles, a greater bond energy needed to be destroyed, so the micro-cutting force was relatively higher.

As shown in Figure 6b, it can be seen that the three potential energy change curves show a trend of first increasing and then gradually reaching a more stable trend, and the time steps to reach the steady state were all about 14,000. Furthermore, the larger the abrasive particles were, the greater the potential energy was. The steady-state values were −41,335 ev, −41,345 ev and −41,350 ev, respectively. Because of the larger particle size of the abrasive particles, more atoms were in contact, so more atomic bonds needed to be broken during the micro-cutting, thereby increasing the potential energy between atoms. This paper also found that the larger the particle size was, the smaller the fluctuation range of the potential energy was, which indicates that the energy system of micro-cutting the workpiece with a large particle size was more stable and more conducive to the workpiece surface formation.

As shown in Figure 6a, according to the change trend of the three curves, with the increase in the particle size, the temperature in the Newtonian zone increased slowly, and the temperature in the steady state of the micro-cutting decreased. However, the overall trend still increased with the increase in particle size, and the temperature in the Newtonian zone with an abrasive particle radius of 12 Å reached 380 K. Moreover, the larger the particle size, the faster the temperature enters the steady state. In this case, the larger the particle size was, the smaller the curvature was, so the degree of plastic deformation of the workpiece formed during the abrasive micro-cutting was reduced. Moreover, the frictional resistance generated by the chips flowing through the front of the abrasive grains was also reduced. Therefore, abrasive grains with a large particle size are more likely to cut into the sapphire workpiece, resulting in faster stable micro-cutting.

### 4.2. Analysis of the Experimental Results

In the experiment, a c-direction (0001) single crystal sapphire slice with a diameter of 50.8 mm and a thickness of 0.5 mm was used. The grinding process was carried out on an intelligent nano-polishing machine system (Nanopoli-100) [31]. The polishing pad was a hydrophilic fixed abrasive polishing pad, and the average particle size of the diamond abrasive were 15 μm, 25 μm, 35 μm, 45 μm, 55 μm, and 65 μm, respectively. The slurry contained deionized water and a small amount of chemicals. The grinding process parameters are shown in Table 2.

The surface roughness (*Ra*) and surface topography of the workpieces after grinding were measured by a Nano Map500LS 5. The mass of the workpiece before and after processing was obtained by a BS224S (220 g/0.1 mg) precision balance. We used Equation (14) to calculate the material removal rate (*MRR*).
(14)$$MRR=\frac{(M_{0}-M)\times h_{0}}{M_{0}\times t}\times 10^{6}$$
where MRR is the material removal rate, in nm/min; $M_{0}$ and M are the masses of the workpiece before and after machining, respectively, in g; $h_{0}$ is the initial thickness of the workpiece before machining, in m; t is the grinding time, in min.

Figure 7 shows the comparison between the experimental values and the theoretical values of the removal rate of the sapphire using FAP grinding with different diamond abrasive particle sizes. After the grinding experiment, the experimental value of the material removal rate was calculated according to formula (14) and compared with the theoretical value. In order to estimate the magnitude of the K value in Equation (10), it was assumed that the experimental and theoretical values during the grinding with a W15 fixed pad were equal, and the calculated K value was used as the basis for theoretical calculation. According to the figure, with the increase in the diamond abrasive particle size in the FAP, the material removal rate obtained from the experiment also increased correspondingly and shows a non-linear increasing trend similar to the theoretical calculation values. When the sapphire workpiece was ground by the FAP with the same particle size, the theoretical value of the corresponding material removal rate was significantly greater than the experimental value, because the distribution of the diamond abrasive particles exposed on the FAP surface has a certain randomness. The actual exposure height was affected by factors such as the holding force of the substrate and the yield, resulting in a low exposure height, and the average depth of cutting into the workpiece was less than the theoretical maximum cutting depth. At the same time, some diamond abrasive grains with low exposure heights had a limited depth of cutting into the workpiece, which may have only caused scratches on the surface of the workpiece. As a result, the material at the notch extends to the periphery, so-called plastic flow occurs, and the goal of material removal cannot be achieved. The elastic deformation of the sapphire workpiece was neglected in the model.

After the experiment, the surface roughness of the workpiece was measured five times at any part of the sampling length of 1500 μm, and the average value was calculated as the final value of the average workpiece surface roughness. In Figure 8, it can be seen that with the increase in the diamond abrasive particle size on the FAP, the theoretical values and experimental values of the surface roughness of the workpiece after machining maintained an approximate linear increasing trend. For the FAP containing larger abrasive grains, the diamond abrasive grains were exposed to a larger height on the surface. The exposed diamond scratched and plowed the surface of the workpiece more significantly and was more likely to leave deep scratches, material collapse, and broken pits, on the surface of the workpiece, causing the surface roughness of the workpiece to increase. In addition, since the depth of cut of the diamond abrasive grains used in the theoretical calculation was greater than the actual value, the elastic deformation of the sapphire workpiece during the grinding process was ignored in the model. As a result, the experimental value of surface roughness of the sapphire workpiece machined by a fixed diamond abrasive pad with the same particle size was less than the theoretical value.

## 5. Conclusions

In this study, the microscopic behavior of diamond grit in the process of FAP grinding of a sapphire was analyzed by modeling, and the effect of the abrasive particle size on material removal was analyzed. The influence of different abrasive particle sizes on the cutting force, system potential energy, and temperature in the Newtonian zone was simulated by molecular dynamics, and the following conclusions were obtained through experimental verification:

(1) During the micro-cutting process of the material, the micro-cutting force increased due to the increasing number of workpiece atoms in contact with the abrasive particles. Then, after entering the micro-cutting stationary state, the micro-cutting force gradually stabilized and fluctuated continuously. The reason may be related to the lattice reconstruction of the material.

(2) Combined with the analysis of the simulation images, the material removal and surface forming mechanisms of the sapphire micro-cutting are summarized as follows: Under the cutting action of the abrasive particles, the sapphire atomic lattice was obviously deformed and reorganized, and then chips and machined surfaces were formed.

(3) In the simulation, three kinds of abrasive particles with different particle sizes were studied and analyzed. It can be seen from the change in the micro-cutting force that the particle size of the abrasive particles had a significant influence on the micro-cutting force at the cutting stage. The influence of the change in particle size on the cutting potential energy and the temperature in the Newtonian zone is very obvious, and both values increased with the increase in particle size. The change in potential energy and temperature reflects the degree of lattice damage, which directly affected the machining quality of the workpiece. Therefore, large abrasive particles will not only obtain a higher removal rate, but they will also cause greater damage to the workpiece.

(4) The material removal rate of the sapphire workpiece with a fixed abrasive pad increased nonlinearly with the increase in diamond abrasive particles, and the surface roughness increased approximately linearly. There are some differences between the theoretical value and the experimental value, but the change trend is consistent. The experimental results confirm the feasibility of the theoretical model.

## Figures and Tables

**Figure 1 micromachines-13-01322-f001:**
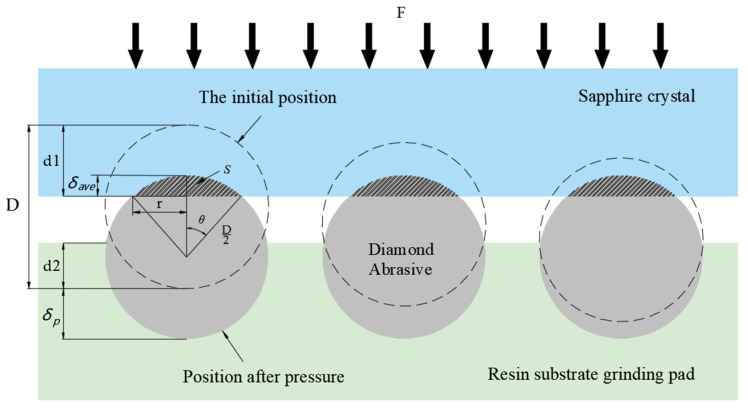
Schematic diagram of abrasive grains cutting into sapphire.

**Figure 2 micromachines-13-01322-f002:**
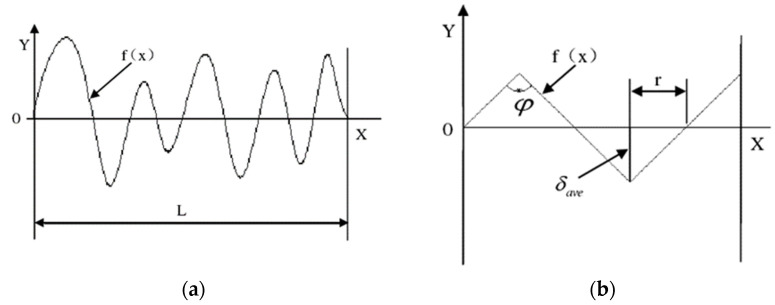
Schematic diagram of surface roughness: (**a**) Surface profile cross section, (**b**) Arithmetic mean deviation micro-element diagram.

**Figure 3 micromachines-13-01322-f003:**
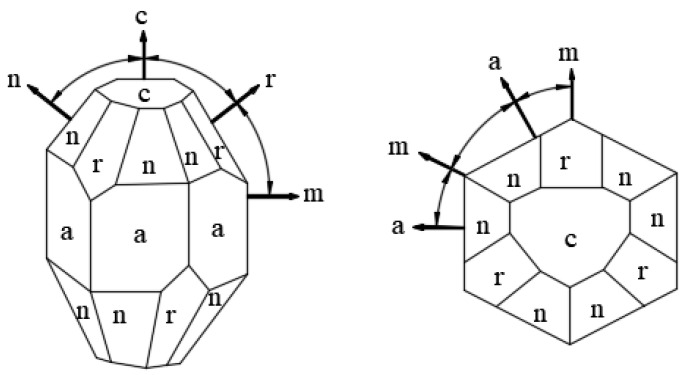
Different crystal planes of sapphire single crystal.

**Figure 4 micromachines-13-01322-f004:**
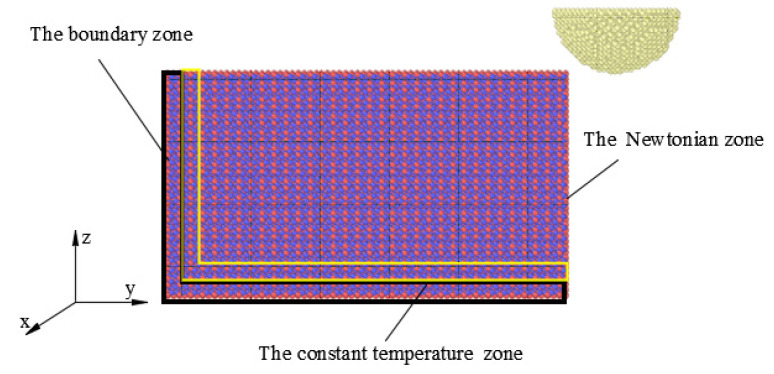
Three-dimensional cutting model.

**Figure 5 micromachines-13-01322-f005:**
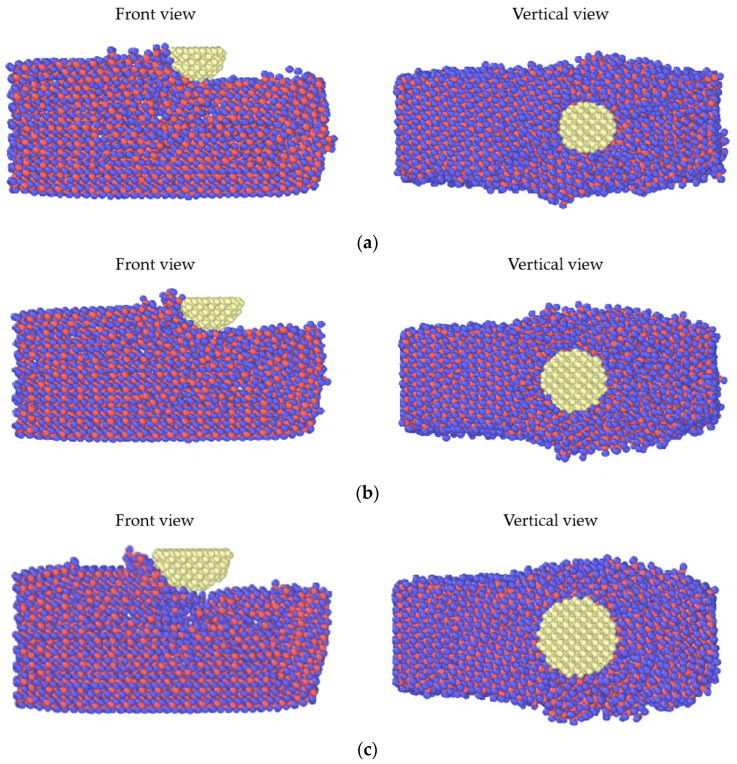
State diagram of micro-cutting simulation under different abrasive radii: (**a**) 8 Å, (**b**) 10 Å, (**c**) 12 Å.

**Figure 6 micromachines-13-01322-f006:**
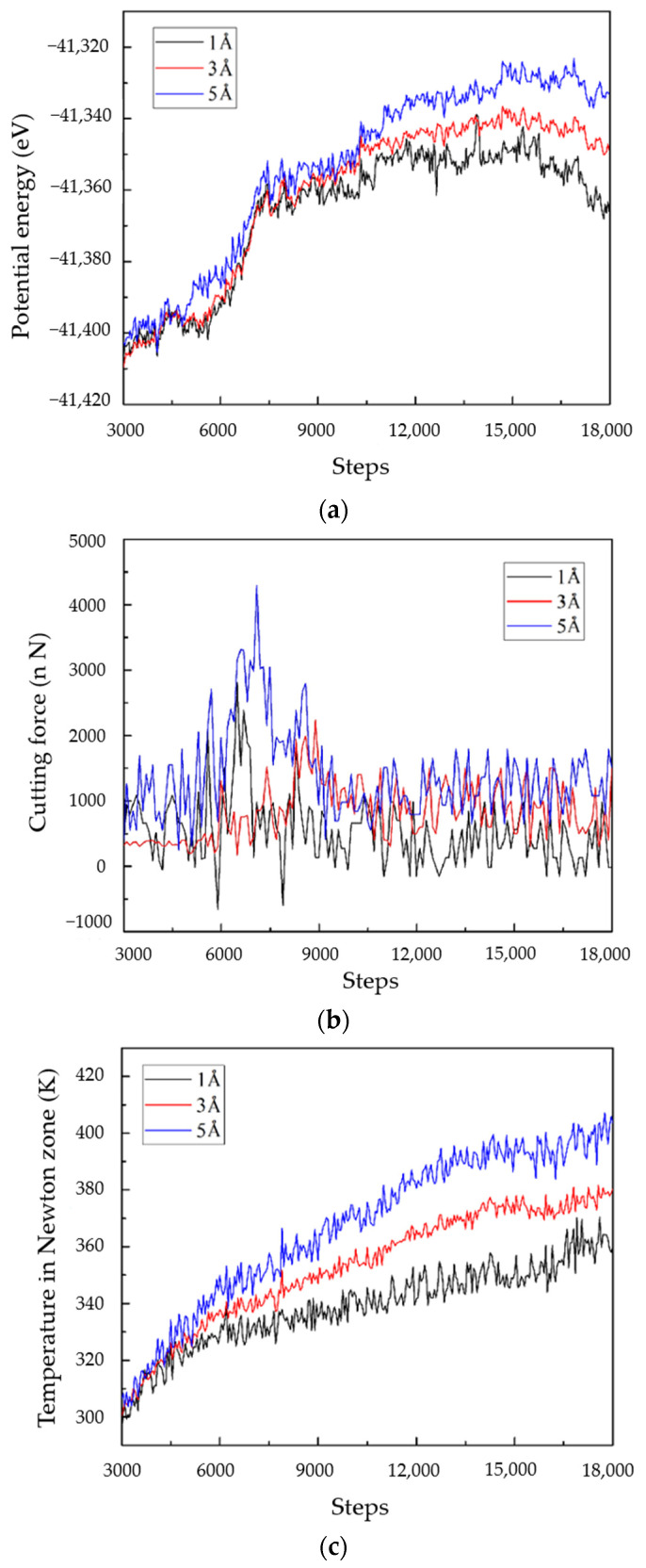
The effect of abrasive particle radius on different physical quantities during the cutting process: (**a**) potential energy, (**b**) cutting force, (**c)** temperature in Newton zone.

**Figure 7 micromachines-13-01322-f007:**
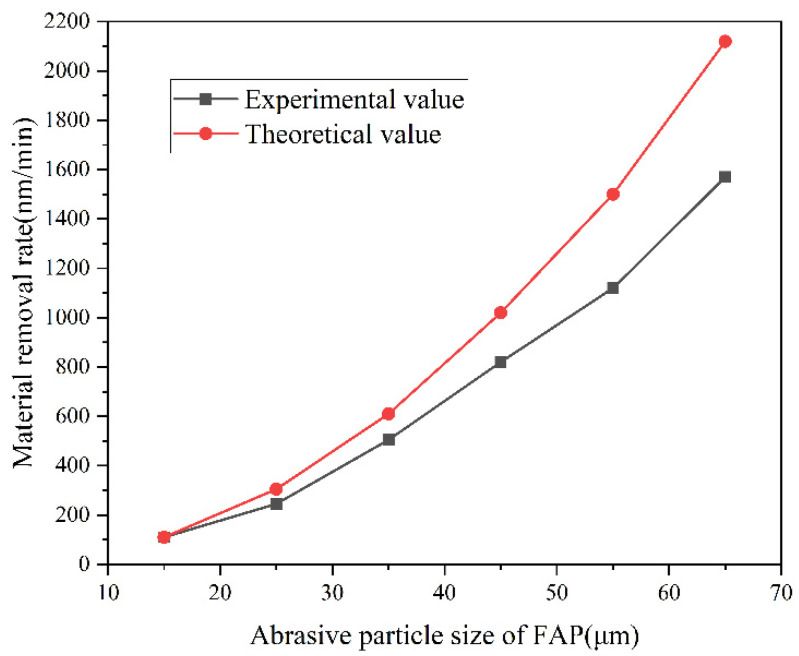
Comparison of experimental and theoretical results of material removal rate.

**Figure 8 micromachines-13-01322-f008:**
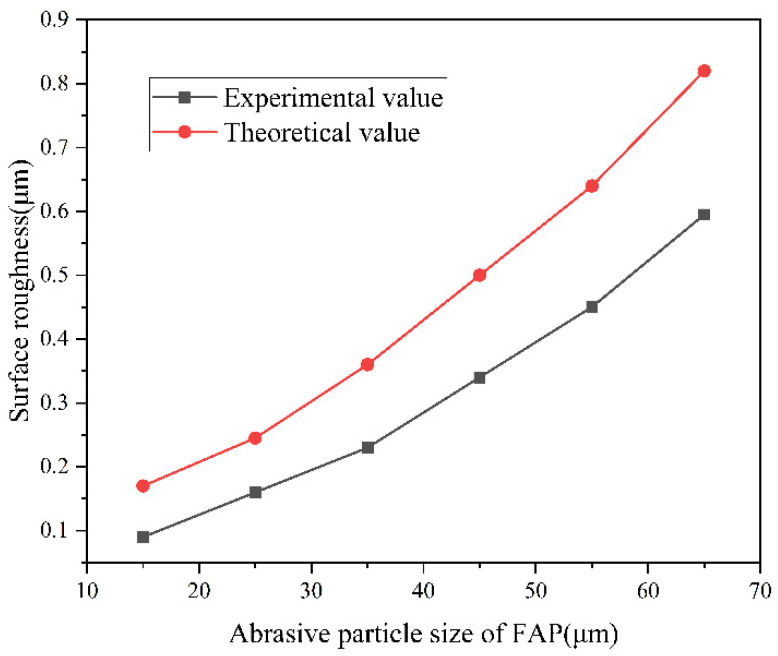
Comparison of experimental and theoretical results of surface roughness.

**Table 1 micromachines-13-01322-t001:** Simulation model parameters of different abrasive particle radii.

Simulation Parameters	Value
Wafer dimensions	90 × 30 × 40 (Å)
Number of wafer atoms	22,004
Cutting speed	150 m/s
Potential functions	Matsui
Initial temperature	300 K
Time step	1 fs
Number of abrasive atoms	2374
polishing depth	R = 5 Å
Abrasive radius	R = 8 Å; 10 Å; 12 Å

**Table 2 micromachines-13-01322-t002:** Processing parameters of grinding experiments.

Parameter	Table Rotation Speed/rpm	Pressure/kPa	Eccentric Distance/mm	Abrasive Fluid Flow Rate mL/min	Time/min
Value	80	34.5	45	100	40

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
