# Peer review of "Effect of Abrasive Grain Concession on Micromechanical Behavior of Lapping Sapphire by FAP"

_micromachines, 2022, doi:10.3390/mi13081322_

Round 1

Reviewer 1 Report

This work explored the material removal mechanism of sapphire at the microscopic level by diamond abrasive grains during the consolidation lapping process in detail. This work can provide some parameters for sapphire consolidation and grinding as reference. I believe this research delivers very solid research on the topic of abrasive grain concession behavior on the sapphire, from both the modeling and experiment. Therefore, this manuscript can be accepted after addressing the following issues.

First, in the introduction part, the author needs to provide more details about sapphire, especially its crystal structure and other structural features. Second, in the simulation, how the atomic model was built? What is the crystal orientation of the upper surface? The modeling should be consistent with the experimental part ({0001} sapphire wafer). Third, I suggest the authors examine the Abstract carefully to avoid unnecessary errors. 

Author Response

Responses to Reviewer #1:

  1. In the introduction part, the author needs to provide more details about sapphire, especially its crystal structure and other structural features?

Reply: Thank you very much for pointing out the question. The chemical composition of single-crystal sapphire is alumina(a-Al2O3). Its constituent elements Al3+ and O2- belong to the typical hexagonal close-packed arrangement; Al3+ ions are in the densely packed O2- inter-octahedral gap, occupying about two-thirds of the gap. The three O2- ions in the sapphire molecule are distributed in the same layer, while the Al3+ ions are alternately arranged in pairs with the O2- ion layer. Every three layers of O2- ions are repeated, keeping the ratio of coordination numbers to O2- ions at 2:3. Due to the stable coordination polyhedral structure, the cleavage and slip of single-crystal sapphire at room temperature are very difficult to activate. I have marked lines 26 to 34 in red font in the text. thank you very much for your suggestion!

  1. In the simulation, how the atomic model was built?

Reply: Thank you very much for your question. it built a model through the lattice custom command in the LAMMPS custom modeling method. The lattice constant of single-crystal alumina was entered (a=4.78 Å, c=12.9914 Å). The length of the basis vector was determined according to the crystallographic information parameters of alumina. Then, the atomic coordinates of Al and O were entered. The atoms were fill into the box. The relative atomic masses of Al and O were defined, and then the atomic model was established. I have marked lines 251 to 257 in red font in the text.

  1. What is the crystal orientation of the upper surface? The modeling should be consistent with the experimental part ({0001} sapphire wafer).

Reply: Due to the special lattice structure of single-crystal sapphire, the bond length and the bond energy between ions are obviously different, resulting in the difference between the energies of each face, which is typical anisotropy. The crystal planes known so far are: c (0001), r (10 1), a ( ), m ( ), n ( ), p ( ), R ( ), s ( ), etc. As shown in figure (Xu, W.H.;Lu, X.C.;Pan, G.S.;Lei, Y.Z.;Luo, J.B. Ultrasonic flexural vibration assisted chemical mechanical polishing for sapphire substrate. Applied Surface Science 2010, 256, 12, 3936-3940), the different faces have different angles. Among them, the surface energy of the c-plane of the single-crystal sapphire is relatively low, and the equivalent coefficient of slip between the crystal basal planes is low. It is easier to achieve a certain plastic deformation through slip. At present, in regarding sapphire crystal orientation, the c-plane is the most widely used (Dobrovinskaya, E.R.;Lytvynov, L.A.;Pishchik, V., Properties of sapphire. Sapphire, Springer, 2009, pp. 55-176). The sapphire workpieces discussed in this article all refer to single-crystal sapphire with face c (0001). I have marked lines 234 to 243 in red font in the text.

Fig Different crystal planes of sapphire single crystal

  1. I suggest the authors examine the Abstract carefully to avoid unnecessary errors

Reply: Thanks for the reviewer's suggestion. We have checked the abstract carefully, and necessary changes have been made in the revised manuscript as far as we can.

Thanks very much for Reviewer 1’s valuable comments. We have improved the manuscript very carefully according to your suggestions.

Reviewer 2 Report

Dear Authors,

Your paper presents too much typos, bonded words, repeated sentences and ideas and errors to be considered for reviewing.

Thius, in my opinion, the paper needs to be firstly revised and, after that, resubmited to be reviewed.

Kind regards.

Author Response

Responses to Reviewer #2:

  1. Your paper presents too much typos, bonded words, repeated sentences and ideas and errors to be considered for reviewing.

Reply: Your suggestion was so meaningful and we appreciate it very much. We are sorry for the major language problems in the article. To address the issue of redaction, the language structure was carefully discussed by the co-authors and significant revisions were made. Furthermore, the article was reviewed by a professional retouching agency. Thank you again for the propose.

Thanks very much for Reviewer 2’s valuable comments. We have improved the manuscript very carefully according to your suggestions.

Round 2

Reviewer 2 Report

Dear Authors,

Thank you for addressing the Reviewers' comments and suggestions.

However, your paper needs a second iteration:

1. Please  revise the English by a English Native Speaker. Some sentences in red colour present several grammatical errors.

2. The gap found in the literature that this paper intends to fill is not clear yet.

3. The results trend is at the end of the Abstract, but if you could add some quantitative results, it would be better for the reader to find interest in reading your paper in full.

Kind regards.

Author Response

  1. Please revise the English by a English Native Speaker. Some sentences in red colour present several grammatical errors.

Reply: Thanks for the reviewer's suggestion. We have checked the article in red colour carefully, and necessary changes have been made in the revised manuscript as far as we can. Thank you very much for your suggestion!

  1. The gap found in the literature that this paper intends to fill is not clear yet.

Reply: Thank you very much for your question. Molecular dynamics (MD) simulation was proven to be a strong candidate for studying the mechanism of FAP removal at the nanoscale, However, there are few literature reports on the material removal and surface forming mechanism of FAP grinding sapphire micro-cutting by using molecular dynamics simulation. I have marked lines 80 to 83 in red font in the text. Thank you very much for your suggestion!

  1. The results trend is at the end of the Abstract, but if you could add some quantitative results, it would be better for the reader to find interest in reading your paper in full.

Reply: Thank you very much for pointing out the question. The simulation results show that the abrasive particle radius was 12Å, the micro-cutting force reached more than 3500nN, while the cutting force with an abrasive particle radius of 8Å only reached 1000nN. I have marked lines 12 to 14 in red font in the text. Thank you very much for your suggestion!

Thanks very much for Reviewer’s valuable comments. We have improved the manuscript very carefully according to your suggestions.
